# A Role for Inositol Pyrophosphates in the Metabolic Adaptations to Low Phosphate in *Arabidopsis*

**DOI:** 10.3390/metabo11090601

**Published:** 2021-09-04

**Authors:** Eric S. Land, Caitlin A. Cridland, Branch Craige, Anna Dye, Sherry B. Hildreth, Rich F. Helm, Glenda E. Gillaspy, Imara Y. Perera

**Affiliations:** 1Department of Plant and Microbial Biology, North Carolina State University, Raleigh, NC 27695, USA; esland@ncsu.edu (E.S.L.); aedye@ncsu.edu (A.D.); 2Department of Biochemistry, Virginia Polytechnic and State University, Blacksburg, VA 24061, USA; cridc2@vt.edu (C.A.C.); craigeb@vt.edu (B.C.); shildret@vt.edu (S.B.H.); helmrf@vt.edu (R.F.H.); gillaspy@vt.edu (G.E.G.)

**Keywords:** inositol pyrophosphates, phosphate starvation response, phosphate homeostasis, lipid remodeling, *AtVIP1*, *AtVIP2*

## Abstract

Phosphate is a major plant macronutrient and low phosphate availability severely limits global crop productivity. In *Arabidopsis*, a key regulator of the transcriptional response to low phosphate, phosphate starvation response 1 (PHR1), is modulated by a class of signaling molecules called inositol pyrophosphates (PP-InsPs). Two closely related diphosphoinositol pentakisphosphate enzymes (*AtVIP1* and *AtVIP2*) are responsible for the synthesis and turnover of InsP_8_, the most implicated molecule. This study is focused on characterizing *Arabidopsis vip1*/*vip2* double mutants and their response to low phosphate. We present evidence that both local and systemic responses to phosphate limitation are dampened in the *vip1*/*vip2* mutants as compared to wild-type plants. Specifically, we demonstrate that under Pi-limiting conditions, the *vip1*/*vip*2 mutants have shorter root hairs and lateral roots, less accumulation of anthocyanin and less accumulation of sulfolipids and galactolipids. However, phosphate starvation response (PSR) gene expression is unaffected. Interestingly, many of these phenotypes are opposite to those exhibited by other mutants with defects in the PP-InsP synthesis pathway. Our results provide insight on the nexus between inositol phosphates and pyrophosphates involved in complex regulatory mechanisms underpinning phosphate homeostasis in plants.

## 1. Introduction

Inorganic phosphate (Pi) is a critical macronutrient required for optimal plant growth and development. Pi uptake by plants is a major limit to crop productivity, and modern agriculture relies heavily on the use of phosphate fertilizers. However, this practice is unsustainable from both geopolitical and environmental perspectives. Currently manufacture of phosphatic fertilizers is dependent on mined rock phosphate and economically viable sources of these minerals are estimated to be depleted within the next 100 years [1]. Compounding this problem, phosphate fertilizers are poorly bioavailable and over-application causes significant ecological damage to fresh and marine waterways. To meet increasing food demand, global agricultural practices must undergo “sustainable intensification” [2]. It is therefore critical that basic research examine the dynamic mechanisms by which plants regulate phosphate uptake, use, and homeostasis.

Plants have evolved several strategies to grow in phosphate-limited soils. Broadly, these adaptive strategies may be divided into two categories: local and systemic. Local adaptations occur in the root and serve to increase Pi uptake from the soil. In tandem, systemic responses aim to maximize Pi usage efficiency and maintain Pi homeostasis within the plant. Local phosphate starvation responses (PSR) are characterized by changes in root system architecture (RSA) which are directly triggered by limiting external Pi [3]. The initiation and elongation of lateral roots redistributes new growth more broadly across the topsoil, while enhanced production of root hairs effectively increases total root surface area and the ability to penetrate into finer soil structures [4]. These adaptations prioritize topsoil nutrient foraging over deep soil water uptake [5]. Systemic or long distance adaptations are responsive to internal Pi concentrations and include reduced photosynthetic activity, metabolic bypasses of Pi-intensive pathways, increased anthocyanin accumulation, membrane lipid remodeling, and increased Pi recovery and transport. These measures are aimed to conserve and recycle phosphate [6,7]. Several mobile signals such as Pi, sugars, hormones, and miRNAs are involved in the coordination between local and long distance signaling pathways [6,8,9,10].

One characteristic long term response which occurs in both shoots and roots under Pi starvation is membrane lipid remodeling [11]. Lipid remodeling involves the concomitant induction of the phospholipid hydrolysis pathway, and the non-phosphate containing glycerolipid biosynthetic pathways [12,13,14,15]. Under low-Pi conditions, glycerolipids such as digalactosyldiacylglycerol (DGDG) and sulfoquinovosyldiacylglycerol (SQDG) replace phospholipids in the plasma membrane, and chloroplast, respectively, to maintain structure and charge across lipid membranes [11,12,16]. Lipid remodeling allows the plant to utilize the approximately ~30% of cellular organic phosphate contained in phospholipids for other essential physiological processes [11,17,18,19].

Underlying the physiological responses to Pi starvation is a massive reprogramming of transcription by which ~10% of the *Arabidopsis* transcriptome may be differentially expressed in response to low Pi. Much of this transcriptional regulation is controlled by the transcription factor, phosphate starvation response 1 (PHR1) [20]. Considered a master regulator, PHR1 modulates the transcription of PSR genes by targeting PHR1 binding sites (P1BS) in their promoter sequences [21]. While PHR1 is not induced by low phosphate, its DNA-binding activity is inhibited in a phosphate-dependent manner by interaction with the nuclear regulatory protein SPX1, (which is named after proteins from *Saccharomyces cerevisiae*, Syg1 and Pho81, and the mammalian Xpr1.) [22]. Recently the mechanism of this regulation was further elucidated with the finding that an electropositive binding pocket conserved in SPX domain structures binds, with increasing affinity, to Pi, inositol phosphates (InsPs), and inositol pyrophosphates (PP-InsPs) [23]. PP-InsPs promote the association of SPX1 with PHR1, thereby inhibiting DNA-binding activity and leading to repression of PSR genes when cellular Pi levels are restored. The high affinity interaction between SPX1 and PP-InsPs, provides a sensitive regulatory mechanism which may be rapidly modulated by relatively small changes in PP-InsP pools [24,25].

Inositol hexakisphosphate (InsP_6_ or phytic acid) serves as the primary phosphate and energy store in most plant seeds [26]. Synthesis of InsP_6_ is catalyzed by the enzyme inositol phosphate kinase-1 (IPK1). Two groups of enzymes are responsible for sequential phosphorylation of InsP_6_ to InP_8_. A multifunctional enzyme, inositol tetrakisphosphate 1-kinase 1 (ITPK1), converts InsP_6_ to InsP_7_ [27,28]. The diphosphoinositol pentakisphosphate (PPIP5K) enzymes referred to as VIP or VIH kinases [29,30] catalyze InsP_8_ synthesis [27,30,31]. A simple pathway of these reactions is shown in Appendix A.

As is evident from several studies, InsPs and PPx-InsPs are implicated in phosphate storage and homeostasis. Disruption of IPK1 leads to over-accumulation of phosphate in shoots and Pi toxicity [32], and activation of PSR genes under Pi-replete conditions [33,34]. Similar to *ipk1*, the *itpk1* mutant also hyper-accumulates Pi in the shoot and shows basally elevated PSR gene expression [34]. The VIP proteins are dual domain enzymes containing an N-terminal kinase domain and C-terminal phosphatase domain with opposing catalytic activities [35,36]. Double mutants with disruptions in the kinase domains of both alleles have been reported previously [31,37]. These mutants have an elevation in InsP_7_ and are almost entirely devoid of InsP_8._ Interestingly, these mutants exhibit severe imbalances in Pi homeostasis [31,37], resulting in accumulation of Pi and constitutive PSR gene expression under Pi-replete conditions. This, along with more recent binding studies [38], has led to a model in which InsP_8_ regulates Pi homeostasis by repressing PSR.

In this paper we describe novel *vip* double mutants that have several opposite phenotypes to *ipk1* and the previously reported *vih* mutants. We report on their basal Pi status as well as local and systemic responses to low Pi. In particular, we compare the shoot lipid remodeling response of the *vip* mutant and *ipk1* to low Pi conditions. Analysis of ^3^H inositol labeling profiles reveals significant differences in InsP_7_ and InsP_8_ patterns in these *vip* mutants as compared to previously reported *vih* mutants, which highlights the complex regulation of plant phosphate homeostasis.

## 2. Results

### 2.1. vip Double Mutants Are Characterized by a Reduced Growth Phenotype and Transcriptional Changes

The *Arabidopsis* VIP [29,30] isoforms *AtVIP1* and *AtVIP2* share similar organization at the genomic and transcript levels, and their predicted protein products are 88% identical. To determine the impact of these enzymes’ activity on plant growth and development in the context of phosphate starvation, multiple T-DNA insertional mutants were obtained from the Gabi-Kat, SALK, and SAIL T-DNA insertion collections (Figure 1a). The homozygosity of each T-DNA insertion was confirmed by genomic PCR, and three unique double mutants (*vip1-1*/*vip2-1*, *vip1-1*/*vip2-2*, *vip1-2*/*vip2-2*) were generated by crossing confirmed parental lines. Each double mutant was further propagated through three generations and homozygosity was selected for at each insertional loci. The allelic combinations of these double mutants (which we will refer to as *vip* mutants) and the previously published *vih* double mutants are listed in Appendix A. As indicated in Figure 1a, the *vip1* alleles that we used to generate double mutants have insertions downstream of the kinase domain. It has been reported that *vip1-2* is able to generate a truncated transcript which contains the kinase domain [31], and potentially retains kinase activity. Quantitative reverse transcription PCR (qRT-PCR) using unique gene-specific primers confirmed that the transcriptional abundance of each loci was significantly reduced in *vip* double mutants (Figure 1b). Specifically, the greatest decreases in *AtVIP1* and *AtVIP2* transcript levels were observed in the *vip1-2*/*vip2-2* double mutant, with the *vip1-1*/*vip2-1* and *vip1-1*/*vip2-2* mutants showing lesser reductions. Furthermore, RT-PCR amplification of cDNA derived from these double mutants did not detect full-length *AtVIP1* or *AtVIP2* transcripts (Appendix A), indicating an absence of functional full length VIP1 and VIP2 proteins.

Morphometric analysis of the double *vip* mutants showed reduced seedling size when grown on Murashige and Skoog (MS) agar media (Figure 1c). In particular, *vip1-1*/*vip2-2* and *vip1-2*/*vip2-2* exhibited a ~20% decrease in primary root length compared to wild-type Col-0 (WT) (Figure 1e). Seedlings transplanted to soil also exhibited reduced rosette diameter (Figure 1d) persistent throughout the vegetative growth phase of mature plants. All three double mutants showed significantly reduced rosette area compared to WT (Figure 1f).

### 2.2. Systemic Phosphate Starvation Response Genes Are Downregulated in vip Double Mutants

Morphometric analyses showed small but significant reduction in growth in the *vip* double mutants compared to WT. To gain insight into the molecular basis of these growth perturbations, we conducted a transcriptional profiling of shoot tissue from 10-day-old WT, *vip1-1*/*vip2-1*, *vip1-1*/*vip2-2*, and *vip1-2*/*vip2-2* seedlings using RNA sequencing.

Three biological replicates for each genotype were processed and differential gene expression was analyzed using DESeq2. Consistent with the transcriptional downregulation of *AtVIP1*/*AtVIP2* quantified in Figure 1b, *vip1-1*/*vip2-1*, *vip1-1*/*vip2-2*, and *vip1-2*/*vip2-2* mutants showed a gradation in numbers of DE genes identified. Relative to WT, (*p*-value [P_adj_] ≤ 0.05) 6, 20, and 168 genes were found to be significantly upregulated *in vip1-1*/*vip2-1*, *vip1-1*/*vip2-2*, and *vip1-2*/*vip2-2* double mutants, respectively. Similarly, 24, 37, and 124 genes were found to be significantly downregulated in these mutants.

Interestingly, a conserved subset of 20 genes were identified as being downregulated in the shoot tissues of all three double mutants relative to WT (Figure 2, Table 1). Of these downregulated genes, 15 (75%) DE transcripts are PSR genes. Several of these genes show involvement in lipid metabolism (*PLDZ2*, *MGD2*, *MGD3*, *SQD1*). Other genes in this subset are also associated with phosphate deficiency responses (*SPX1*, *PS2*, *IPS1*, *AT4*, *PAP17*, *GDPD1*). Additionally, three related PSR genes (*SQD2*, *VTC4*, *GDPD6*) were found to be downregulated in *vip1-1*/*vip2-2* and *vip1-2*/*vip2-2* mutants only. A similar downregulation of PSR regulatory genes (*SPX1*, *AT4*, *IPS1*) was observed in the roots of *vip* double mutants when queried by qRT-PCR (Appendix A). To identify potential common cis-acting regulatory elements, the 1 kb region upstream of these genes was examined. Interestingly, one or more PHR1-binding sequences (P1BS, GnATATnC) were identified in 14 of the 20 downregulated PSR genes. The presence and number of P1BS sequences in the 20 common genes is indicated in Table 1. This suggests that the subset of downregulated genes in the *vip* double mutants are likely under the control of the PHR1 transcription factor.

### 2.3. Inorganic Phosphate Levels Are Altered in vip Double Mutants

To examine the consequences of altered PSR transcript abundances on the state of cellular Pi levels in the double *vip* mutants, young seedling and mature leaf tissues were analyzed for Pi content by a molybdate-based colorimetric assay and compared to WT and the *ipk1* mutant which hyperaccumulates phosphate (Figure 3).

In both seedlings and mature leaves, Pi levels were significantly increased in the *ipk1* mutant, relative to WT, consistent with previous reports [32,33,34]. Interestingly, the *vip* double mutant seedlings showed small reductions in phosphate content relative to WT. A gradient of this phenotype was found to be consistent with the previously observed transcriptomic gradient. The *vip1-2*/*vip2-2* double mutant seedlings exhibited the greatest decrease in Pi levels at 18 days old relative to WT (~28%) while *vip1-1*/*vip2-2* exhibited ~12% decrease relative to WT. In mature leaf tissues of the *vip* mutants, Pi levels were similar to those of WT.

### 2.4. Root System Architecture Is Significantly Altered in vip Double Mutants’ Response to Low Phosphate

WT and *vip* double mutants were grown on either replete or limiting phosphate MS media to monitor changes in RSA. As expected, after 14 days of growth on limiting Pi, wild-type roots showed reduction in primary root growth along with increased lateral roots and increased root hairs compared to the Pi-replete condition (Figure 4a–c). In contrast, *vip* double mutants exhibited significantly reduced lateral root formation and root hair elongation under low Pi compared to wild type. Consistent with other phenotypes, *vip1-1*/*vip2-1* mutants showed a ~10% reduction in lateral root formation (Figure 4b). More notably, both *vip1-1*/*vip2-2* and *vip1-2*/*vip2-2* showed dramatic reductions in lateral root formation (~50%). A similar trend was observed with root hair elongation (Figure 4c). Relative to WT, *vip1-1*/*vip2-1* mutants exhibited a ~23% decrease in root hair length while *vip1-1*/*vip2-2* and *vip1-2*/*vip2-2* exhibited a ~40% decrease in root hair length. Together these data show that *vip* mutant seedlings exhibit dampened local responses to low phosphate availability.

### 2.5. Anthocyanin Accumulation Is Reduced in vip Double Mutants in Response to Low Phosphate

Foliar anthocyanin accumulation is associated with a systemic physiological response of plants to low phosphate availability [39]. Interestingly, the transcription factor production of anthocyanin pigment1 (PAP1), a key controller of anthocyanin biosynthesis, was recently shown to be regulated via SPX4 and InsPs, analogous to the PHR1/SPX1 inhibition complex [40,41].

After growing on phosphate-limiting media for 18 days, *vip* double mutant seedlings accumulated considerably less anthocyanin in their shoot tissues, relative to WT controls. (Figure 5a). As measured by a spectrophotometric anthocyanin assay, *vip1-1*/*vip2-1*, *vip1-1*/*vip2-2*, and *vip1-2*/*vip2-2* mutants accumulated 66%, 69%, and 76% less anthocyanin than WT controls. This dramatic reduction in anthocyanin accumulation in response to limited Pi in the *vip* mutants supports a role for PP-InsP-mediated regulation anthocyanin biosynthesis.

### 2.6. Phosphate Starvation Response (PSR) Genes Are Inducible in vip Double Mutants

To query the ability of *vip* double mutants to surmount a systemic phosphate starvation response, WT and *vip* mutant seedlings were grown under replete or limiting phosphate conditions for 18 days and transcriptional responses in the shoot tissues were examined (Figure 5b). The subset of PSR genes found to be basally downregulated in *vip* double mutants (Table 1), along with other genes involved in lipid metabolism, were monitored by qRT-PCR. Consistent with the RNA-seq results, basal expression of many of the queried genes was reduced compared to wild type. However, this downregulation was only seen in a subset of PSR genes; the basal level of expression of *Pht1;5* (a high-affinity phosphate transporter in leaves) was comparable to that of wild type. Irrespective of their basal levels in the *vip* mutants, all PSR genes queried showed significant induction under limited phosphate (Figure 5b). Transcripts associated with lipid metabolism including *PLDZ2*, *MGD2*, *MGD3*, and *SQD2* were induced under low Pi in *vip1-1*/*vip2-1*, *vip1-1*/*vip2-2*, *vip1-2*/*vip2-2*. Similarly, transcripts associated with phosphate transport during leaf senescence, [42,43] including *GDPD1* and *Pht1;5,* were upregulated. Furthermore, the relative fold change under low Pi in *vip* doubles was comparable to PSR induction observed in shoots of WT seedlings, indicating that the *vip* double mutants are equally responsive and not impaired in their transcriptional response to low Pi.

### 2.7. vip1-2/vip2-2 Mutants Are Deficient in InsP_8_

To determine if the changes in these vip mutants were associated with alterations in PP-InsPs, we focused on the *vip1-2*/*vip2-2* double mutant, and quantified PP-InsPs by radiolabeling with ^3^H-myo-inositol using our previously described methods [27]. We also compared *vip1-2*/*vip2-2* mutants to *ipk1*. We found, as expected and as shown by others, that *ipk1* mutants contained elevated Ins(1,3,4,5,6)P_5_, greatly reduced InsP_6_, and a reduction in InsP_7_ and InsP_8_ (Figure 6a,b,e–g). In contrast, *vip1-2*/*vip2-2* mutants exhibited a ~25% decrease in InsP_8_ levels, a slight reduction in InsP_7_ and no significant changes in other InsPs (Figure 6c–g). We conclude that the *vip1-2*/*vip2-2* mutant has a unique alteration in its InsP/PP-InsP profile, which most resembles a loss of function in the VIP kinases.

### 2.8. Altered Lipid Remodeling Responses in PP-InsP-Deficient Mutants

In response to low Pi, plants evoke a well-characterized lipid remodeling process that involves the replacement of phospholipids by galactolipids and sulfolipids within plant membranes [13,14,15]. To examine whether changes in PP-InsPs affect lipid remodeling, we assessed the response of *vip1-2*/*vip2-2* and *ipk1* mutants relative to WT using an untargeted mass spectrometry approach.

As seen in by the PLS-DA plot (Appendix A), there are dramatic differences between Pi-replete and Pi-limited conditions in shoot tissues of WT and both *vip* and *ipk1* mutants (with the treatment variable contributing to >22% of the difference). Additionally, all of the replicates for each genotype/treatment clustered together. We conclude that our MS conditions are appropriate to measure established changes in lipid remodeling in response to low Pi.

We investigated the levels of the different classes of phospholipids and galactolipids under normal growth conditions, and in response to low Pi. WT plants responded to low Pi conditions as expected. Specifically, our data indicate that various species of phosphatidylcholine (PC), phosphatidylethanolamine (PE), phosphatidylglycerol (PG), phosphatidic acid (PA), and phosphatidylinositol (PI) undergo a significant decrease (Figure 7, Appendix A). Conversely, various species of DGDG, SQDG, and triacylglycerol (TAG) significantly increase in response to low Pi, while monogalactosyldiacylglycerol (MGDG) levels do not change (Figure 7, Appendix A).

We compared the *vip* mutant with WT plants under both Pi-replete and low-Pi conditions. The *vip1-2*/*vip2-2* mutants grown under Pi-replete conditions did not significantly differ in any lipids queried (Figure 7, Appendix A) compared to WT. When *vip1-2*/*vip2-2* mutants were grown under low-Pi conditions, levels of DGDG, SQDG, and TAGs increased relative to the Pi-replete condition. However, the levels of these lipids were significantly lower than in similarly grown WT plants (Figure 7, Appendix A). We conclude that *vip1-2*/*vip2-2* mutants have suppressed a portion of their low-Pi lipid remodeling responses, specifically with regards to DGDG, SQDG, and TAGs.

We included *ipk1* mutants in our analysis as these plants have known changes in both InsP_6_ and PP-InsPs, which we confirmed in Figure 6. The data presented in Figure 7, Appendix A indicate that *ipk1* mutants differ significantly from WT plants grown in replete-Pi conditions in their levels of both phospholipids (PC, PG, PI) as well as galactolipids (MGDG, DGDG) and sulfolipids (SQDG). As each of these lipid classes is significantly lower in *ipk1* mutants, we conclude that under Pi-replete conditions, altered levels of InsP_6_ and PP-InsPs correlate with suppressed levels of key lipids regulated by the phosphate sensing pathway. Interestingly, under low Pi, decreases in phospholipids (PG, PI) and increases in DGDG and SQDG in *ipk1* are comparable to those in WT. Thus, we conclude that lipid remodeling in response to low Pi is not impaired in *ipk1*.

When comparing the lipid levels and lipid remodeling responses of *vip1-2*/*vip2-2* and *ipk1* mutants, our data support a model in which IPK1 preferentially impacts lipids under Pi-replete conditions, while the VIP enzymes have a greater impact on lipid remodeling in response to low Pi.

## 3. Discussion

### 3.1. Comparison of Mutants with Disrupted Inositol Pyrophosphate Biosynthesis

In this paper we characterize novel *vip* double mutants and compare their responses to limited Pi. Specifically, we show that both local (RSA) and systemic (metabolic) responses to low phosphate are dampened/reduced in the *vip* mutants compared to wild type. These mutants provide an important tool for discerning the molecular details of Pi-sensing in plants as they contain novel changes in PP-InsPs, which are implicated in the repression of PSR. Specifically, the *vip* mutants differ in their PP-InsP profiles from previously published *vih* mutants [31,37]. We show that the *vip* mutants contain a modest reduction in InsP_8_ (70–75% of WT levels), as detected by ^3^H *myo*-inositol labeling (Figure 6c,d,g). Furthermore, our *vip* mutants differ significantly in the amount of InsP_8_ precursors, i.e., InsP_6_ and InsP_7_, as compared to *ipk1* and *vih* mutants. [31,37]. Notably, the differences in the levels of InsP_7_ and InsP_8_ between the *vip* and the *vih* mutants result in near opposite phenotypes, with respect to the regulation of Pi homeostasis and PSR, including gene expression (Figure 5b), Pi accumulation (Figure 3), and anthocyanin production (Figure 5a). Importantly, the transcriptomic data show that under Pi-replete conditions a subset of PSR genes are repressed in shoots and roots of *vip* mutants compared to WT levels. This is in contrast to other mutants with lower InsP_8_ levels, including both the *vih* and *ipk1* mutants. This suggests two important considerations for Pi-sensing. Firstly, our data cannot be explained by the current model of PHR1 regulation. In this model, InsP_8_ is thought to be the primary regulatory ligand promoting repression of PHR1 by SPX1; in the absence of InsP_8_, PSR gene expression is constitutively upregulated under Pi-replete conditions. However, in the *vip* mutants, PSR gene expression is not constitutive despite a reduction in InsP_8_; in fact, their response is opposite. Given that SPX domains are able to interact with less phosphorylated forms (InsP_6_ and InsP_7_), and that the cellular levels of these metabolites may far exceed that of InsP_8_, it is conceivable that other InsPs and PP-InsPs may contribute towards the regulation of PSR. The second novel implication from our transcriptomic analysis is that only a subset of PSR genes is misregulated in the *vip* mutants. This suggests that specific subsets of genes, and possibly metabolites as well, are regulated by a finely tuned InsP and/or PP-InsP profile within the plant.

### 3.2. Transcriptional Responses of vip Mutants

A stated above, it is intriguing that we identified downregulation of 20 PSR genes in all three of the *vip* mutants reported here. This is all the more striking given that each of the three mutants have overlapping but different allelic combinations. Several of these (14) genes were also shown to contain a variant of the P1BS element in their promoters that is presumably targeted by PHR1. Furthermore, these genes belong to a *PLDz* co-expression network [44], which includes genes involved in lipid remodeling and Pi recycling. This suggests that additional factors affecting PHR1 DNA-binding may be required to provide transcriptional specificity.

We also compared the transcriptional results from *vip1-2*/*vip2-2* with those of *vih1-2*/*2-3,* a double mutant with a different allelic combination [37]. A major difference in differentially expressed genes (DEGs) was that the 20 basally downregulated genes in *vip1-2*/*vip2-2* were upregulated in *vih1-2*/*2-3.* Despite this difference, we also detected substantial overlap of DEGs between the two mutants (Appendix A). In fact, 57% of upregulated and 47% of downregulated genes in *vip1-2*/*vip2-2* are shared with *vih1-2*/*vih2-3.*

Gene ontology (GO) terms enriched in the upregulated genes include stress response genes, as well as those involved in immune responses and ethylene-activated signaling pathways. Compared to wild type, several leucine-rich repeat (LRR)-receptor-like protein kinases, WRKY transcription factors, ethylene responsive transcription factors, UDP-glycosyl transferases, as well as JAZ1 (a negative regulator of Jasmonic Acid (JA) signaling) were upregulated in both *vip1-2*/*vip2-2* and *vih1-2*/*2-3* mutants. Some of these defense-related genes were also found to be induced within 1 h in response to jasmonic acid (JA) treatment [45,46]. Interestingly, InsP_8_ levels were shown to increase in *Arabidopsis* leaves upon treatment with methyl jasmonate and it was proposed that InsP_8_ and JA bind the COI1-ASK1 complex functioning as an incidence signal to recruit the JAZ1 repressor protein for ubiquitin-mediated degradation [30], thereby activating JA-mediated defense responses. Single *vih2-3* or *vih2-4* mutants had reduced expression of JA response genes *VSP2* and *MYC2* and were more susceptible to herbivory [30]. Given the response of the single mutants, it is intriguing that both the *vip1-2*/*vip2-2* and the *vih1-2*/*vih2-3* double mutants show increased expression of *JAZ1* and other JA-responsive genes basally. In fact, a closer examination of the *vih1-2*/*vih2-3* double mutant DEG dataset indicates that several genes involved in JA biosynthesis (*AOS*, *AOC*, *LOX2*, *LOX3*, *LOX4*, and *LOX6*), as well as the JA responsive master regulator *MYC2* [47] and the vegetative storage proteins *VSP1* and *VSP2,* have greater than twofold increased basal expression [37]. Taken together, these observations suggest that some aspects of defense signaling may be under the control of compensatory pathways. It is conceivable that ethylene crosstalk may play a role, since JA-ET can act in an interdependent manner to control plant defense [48].

Gene ontology (GO) terms enriched in the downregulated genes are primarily associated with the chloroplast and include several genes encoding for subunits of the photosynthetic apparatus, including photosystem I (PS I), components of the oxygen evolving complex, chlorophyll ab-binding protein subunits, as well as NADPH quinone oxidoreductase, and the chloroplastic NDH. The NDH complex is tightly associated with PS I and functions in cyclic electron transport [49]. It is remarkable that approximately 50% of the downregulated genes identified in the *vip* mutant are associated with the photosynthetic apparatus, yet this does not appear to dramatically impact overall growth. The connection between photosynthesis and PP-InsPs thus awaits a thorough analysis of photosynthetic parameters within *vip* and *vih* double mutants.

### 3.3. Metabolite Changes in vip Mutants

Given that we identified transcriptional changes in genes that encode enzymes involved in lipid remodeling, we used LC-MS to profile key lipid changes in *vip* double and *ipk1* mutants. Our expectation was that analysis of these two mutants, one with suppressed PSR gene expression and one noted for increasing PSR gene expression, would give us a chance to discern how PP-InsP-induced transcriptional changes impact lipid remodeling in response to low Pi. Under limited Pi, we expected to detect increases in galactolipids and sulfolipids in *vip* mutants comparable to those in WT, given their robust transcriptional responses to limiting Pi (Figure 5b). However, we observed an attenuated response of DGDG and SQDG increases under limited Pi. These results reveal two important insights. The first is that small changes in PP-InsPs can alter lipid remodeling in response to low Pi conditions. The second is that this altered regulation of lipid remodeling in response to low Pi is not controlled at the transcriptional level in the *vip* mutant, suggesting that there are other, post-transcriptional mechanisms involved.

Many researchers have characterized adaptations of the *ipk1* mutant to low phosphate [33,34]. However, to our knowledge, lipid remodeling has not been examined. If *ipk1* shoots are responding to a perceived Pi limitation, we might expect decreases in phospholipids with concomitant increases in galactolipids and sulfolipids under basal Pi conditions. Instead, we observed a significant reduction in PC and PG as well as in the galactolipids DGDG and SQDG (Figure 7a–d). These results indicate that replete Pi conditions may stimulate hydrolysis of specific phospholipids in the *ipk1* mutant, however, this does not result in an increase in galactolipids and sulfolipids. Under limited Pi, *ipk1* mutants contain similar levels of PC, PE, DGDG, and SQDG as compared to WT. This indicates that despite the changes in lipid composition under replete Pi conditions, *ipk1* is able to turn on a robust lipid remodeling response.

Under Pi stress, *Arabidopsis* leaves were shown to accumulate up to ~20-fold greater TAG [19]. In *Chlamydomonas reinhardtii*, mutation of *VIP* abrogates control of nutrient sensing, and results in an elevation of TAGs under replete Pi and under nitrogen starvation [50]. We found that *vip* mutants exhibited similar levels of TAG to WT under Pi replete conditions. However, under limited Pi, TAG levels were significantly reduced in *vip* mutants (Figure 7e), further supporting that a dampened response to low Pi occurs in these *vip* mutants. In contrast, *ipk1* exhibited a small, but insignificant, reduction in TAG levels under both Pi-replete and Pi-limiting conditions. These data could inform engineering strategies to increase TAG production in higher plants.

Taken together, the expression patterns of genes involved in phospholipid turnover and galactolipid and sulfolipid synthesis do not correlate directly with the lipid profiles for the mutants studied here. This incongruence indicates that there must be a higher order of control of lipid remodeling processes, beyond what occurs at the regulation of transcription of lipid remodeling genes.

## 4. Materials and Methods

### 4.1. Plant Materials and Growth Conditions

Seed of T-DNA insertional mutants were obtained from the Gabi-Kat (*vip1-1* GK_204E06; *vip1-2* GK_008H11), SALK (*vip2-1* SALK_094780), and SAIL (*vip2-2* SAIL_175_H09) T-DNA insertion collections [51,52,53]. Double mutants generated from these insertional lines were compared to Col-0 WT in all experiments. In all experiments carried out on Murashige and Skoog (MS) agar media, seeds were surface sterilized in a dilute bleach solution (2% *v*/*v* sodium hypochlorite, 0.1% *v*/*v* Triton X-100, aq.) for 12 min and rinsed with eight changes of sterile deionized water. Seeds were stratified at 4 °C in darkness for 2 days and sown on plates containing 0.5× MS media (M576, Phytotech Labs, Lenexa, KS, USA) supplemented with 1% *w*/*v* sucrose, 0.05% morpholinoethanesulfonic acid, 1% *w*/*v* agar (Agar-M, Sigma Aldrich, St. Louis, MO, USA), pH 5.7. Seedlings were grown under an 8 h light/16 h darkness photoperiod with an intensity of 110 µmol m^−2^ s^−1^. Day/night temperatures were set to 21 °C/19 °C.

### 4.2. Basal Morphometric Analysis

*Arabidopsis* seedlings were grown as described above and primary root length was measured after 10 days using morphometric image analysis software (ImageJ v1.52n). Eighteen-day-old seedlings were transplanted to wetted soil (PGX, Pro-Mix) and mature rosette leaf areas were measured after 28 days of growth.

### 4.3. Low Phosphate Treatment

For all low-phosphate experiments, phosphate-replete media contained 1 mM Pi and phosphate-limiting media contained 10 µM Pi. Media were prepared by supplementing commercially prepared MS micronutrient media (M529, Phytotech) with macronutrients (20.61 mM ammonium nitrate, 2.99 mM calcium chloride dihydrate, 3.07 mM magnesium sulfate heptahydrate, 18.79 mM potassium nitrate) and phosphate in the form of monobasic potassium phosphate. All glassware used for low-phosphate experiments was rinsed for 5 min in a solution of 0.1 M NaOH/0.1 M HCl, and rinsed 10 times with deionized water to remove accumulated phosphate from the borosilicate glass.

WT and *vip* double mutant seedlings were grown on 1× MS medium with replete (1 mM Pi) or limiting (10 µM Pi) phosphate. For all low-phosphate treatment experiments, plates were placed within the growth chamber in a random arrangement by genotype and treatment with regular rearrangement to minimize any effects of microenvironments within the chamber. Roots were imaged after 14 days under a stereo microscope (Wild M3, Heerbrugg, fitted with MC170HD camera, Leica, Wetzlar, Germany), and root hairs growing within the focal plane (orthogonal to the axis of observation) were measured using morphometric image analysis software (ImageJ v1.52n).

### 4.4. RNA Isolation and NextGen RNA Sequencing

WT, *vip1-1*/*vip2-1, vip1-1*/*vip2-2, vip1-2*/*vip2-2* seedlings were grown on 0.5× MS media under standard conditions as described above. Shoot tissues (all aerial tissues including leaves, cotyledons, and hypocotyls) were harvested after 10 days of growth. RNA was isolated using a silica column-based kit (RNEasy, Qiagen, Hilden, Germany) and analyzed for integrity by capillary electrophoresis (BioAnalyzer 2100). Three replicate experiments were carried out. Directional next generation RNA sequencing libraries were prepared for the Illumina HiSeq platform according to manufacturer’s instruction (NEBNext Ultra directional) and each sample was uniquely barcoded. Samples were analyzed in triplicate across three lanes of an Illumina HiSeq chip carried out at the Genomics Sciences Laboratory (GSL, NCSU). Reads were trimmed by CutAdapt (https://github.com/marcelm/cutadapt/tree/v1.7.1 (accessed on 1 September 2017)) to unconditionally remove the first 10 bases of each read, search for and remove universal Illumina adapter prefixes, and reject reads below 50 bp. Trimmed FASTQ files were aligned to the *Arabidopsis* genome (Ensembl TAIR10 genes.gtf) by the tophat pipeline using bowtie2, resulting in an average mapping rate of 97.5%. Differential gene expression was analyzed by linear modeling using DESeq2 (https://bioconductor.org (accessed on 13 September 2017)). A linear model was built to include genotype and bio-replicate groupings and contrast terms were used to compare basal gene expression of each *vip* double mutant to WT. Differential expression lists were generated to include genes which had >10 reads, and adjusted *p*-values of <0.05. GeneID lists were annotated by functional classification using TAIR and the PANTHER classification system (www.arabidopsis.org, www.pantherdb.org (accessed on 16 September 2017)).

### 4.5. Inorganic Phosphate Assay

WT, *vip1-1*/*vip2-1, vip1-1*/*vip2-2, vip1-2*/*vip2-2,* and *ipk1* seedlings were grown on 0.5× MS media at 21 °C under an 8 h photoperiod for 18 days. Separately, a replicate panel of seedlings were transplanted to soil and grown to maturity for an additional 28 days. Shoot tissues from ~50 seedlings/genotype/bioreplicate were harvested into liquid nitrogen and ground to a fine powder. For assays of mature plants, 2 leaves were collected from each of 8 plants/genotype/bioreplicate and ground in a similar manner. Inorganic phosphate was extracted by incubating in ten volumes 1% *v*/*v* acetic acid. Extracted fractions were analyzed for Pi concentration by a modified micro-titer assay [54]. Briefly, 50 µL sample extract was diluted to a volume of 150 µL, mixed 1:1 with an ammonium molybdate working reagent (5% *w*/*v* FeH_14_O_11_S·7H_2_O, 1% *w*/*v* (NH_4_)_2_MoO_4_, 1N H_2_SO_4_ aq.) and incubated for 1 h. Absorbance at 660 nm was measured on a plate spectrophotometer and phosphate concentrations were calculated using a standard curve of known phosphate amounts in the form of KH_2_PO_4._

### 4.6. Anthocyanin Measurements

Shoot tissue was harvested from 18-day-old seedlings (~50 seedlings/genotype/bioreplicate) grown on Pi-limiting (10 µM Pi) MS medium. Anthocyanin was extracted and quantified by a modified method adapted from [55]. Briefly, 50 mg shoot tissue was homogenized by 2.8 mm stainless steel beads in 500 µL extraction buffer (50% *v*/*v* EtOH ((aq.)) containing 0.25% *v*/*v* HCl) using a tissue disruption apparatus (BioSpec Mini-Beadbeater). Lysates were clarified by centrifugation at 20,000 RCF and 300 µL supernatant was transferred to clean microcentrifuge tubes. Collected supernatants were washed with three changes of 400 µL CHCl_3_ and vigorous shaking. Isolated anthocyanin fractions were drawn from the upper phase, diluted 20-fold and quantified by reading absorbance at 524 nm using a spectrophotometer.

### 4.7. Statistical Analysis of Morphometric and Spectrophotometric Data

For all morphometric analyses, the number of biological and technical replicates is indicated for each experiment in accompanying figure legends. For spectrophotometric data, including inorganic Pi and anthocyanin measurements, 2-3 biological replicates (described in the methods sections above) were analyzed. Morphometric and spectrophotometric data were analyzed by one-way ANOVA using JMP Pro v15 software (SAS Institute Inc., Cary, NC, USA). Post hoc analysis of these data were carried out by Tukey–Kramer honest significant difference tests at a significance level of α = 0.05.

### 4.8. RNA Isolation, cDNA Synthesis, and qRT-PCR Analysis

Shoot tissue was ground to a fine powder in liquid nitrogen and total RNA was isolated using a silica column-based kit (RNEasy, Qiagen, Hilden, Germany). RNA isolates were treated with DNase (DNAfree, Invitrogen, Carlsbad, CA, USA) and first strand cDNA was synthesized with random hexanucleotide primers and Multiscribe reverse transcriptase according to manufacturer’s specifications (Applied Biosystems, San Francisco, CA, USA). Quantitative reverse transcription PCR (qRT-PCR) reactions were prepared with a mastermix (PowerUp SYBRGreen, Applied Biosystems) and carried out in triplicate using cDNA equivalent to 20 ng of RNA input. *AtVIP1* and *AtVIP2* transcript abundances were quantified using gene-specific primers and *PP2A* as an endogenous control. Primer efficiencies for *AtVIP1*, *AtVIP2*, and *PP2A* assays were determined using a standard curve and found to be comparable. PSR gene expression was quantified in low-phosphate-treated samples using gene-specific primers (Appendix A) and *PEX4* as an endogenous control. All qRT-PCR reactions were carried out in a StepOnePlus Thermal cycler (Applied Biosystems) and relative expression values were calculated according to the ΔΔCT method [56] by instrument software (StepOne v2.2.2).

### 4.9. Plant Inositol Polyphosphate Analysis

*Arabidopsis* seedlings were grown in 0.5 × MS + 0.2% *w*/*v* agarose, pH 5.7 with a 16 h light, 8 h dark cycle. After 2 weeks of growth, 50 µCi of [^3^H]-*myo*-Inositol (20 Ci/mmol, American Radiolabeled Chemicals, St. Louis, MO, USA) was added to 24 seedlings in the same media. After 5 days of incubation under constant light, the seedlings were rinsed 3 times with water, frozen, then thawed and extracted with 1 M perchloric acid + 3 mM EDTA + 0.1 mg/mL InsP_6_ (Sigma Aldrich, St. Louis, MO, USA) by douncing with a plastic pestle on ice. The extract was then centrifuged for 5 min at maximum speed, the supernatant was transferred to a microcentrifuge tube and neutralized to pH 6–8 with 1 M potassium carbonate + 3 mM EDTA. The tubes were loosely capped and incubated on ice for 2 h, then centrifuged at maximum speed for 5 min. The supernatant was transferred to a microcentrifuge tube and concentrated under a stream of nitrogen. Inositol polyphosphates were separated by HPLC using a 125 × 4.6 mm Partisphere SAX-column (MAC-MOD Analytical, Chadds Ford, PA, USA), and eluted with a previously described ammonium phosphate gradient [57]. Eighty-five fractions were collected and radioactivity was measured by liquid scintillation counting (Beckman Coulter, New Brunswick, NJ, USA). To exclude unincorporated label, the total CPM of fractions 13–85 was used to normalize the datasets. Normalized CPM from fractions comprising the peaks for InsP_6_, InsP_7_, and InsP_8_ were used for quantification. Error bars are standard deviations of the mean (*n* = 3 biological replicates) and statistical significance values of deviations from WT means were calculated by two-tailed Student’s *t*-test.

### 4.10. Lipidomics Methods

Lipid extraction method was followed according to an amended protocol [58]. WT, *vip1-2*/*vip2-2,* and *ipk1* seedlings were grown on 0.5× MS media at 21 °C under an 8 h photoperiod for 18 days. Six biological replicates of ~50 seedlings/sample were processed for each genotype and treatment. Shoot tissue was harvested into 3 mL vials, containing 0.5 mL isopropanol (Spectrum Chemicals, New Brunswick, NJ, USA) with 0.01% BHT (Sigma-Aldrich, St. Louis, MO, USA) preheated to 75 °C. Samples were incubated at 75 °C for 15 min and cooled to room temperature. 1.5 mL chloroform:methanol:water (30:41.5:3.5), Spectrum Chemicals, New Brunswick, NJ, USA) was added and shaken at 100 rpm on an orbital shaker for 24 h. Before lipid analysis, 70 µL of 600 mM ammonium acetate (Sigma-Aldrich, St. Louis, MO, USA) was added.

Lipid profiling was performed using a Waters Acquity I-class ultra-performance liquid chromatography (UPLC) coupled with a Waters Synapt G2-S Q-TOF mass spectrometer (Waters Corp, Milford, MA, USA). Sample separation was achieved with a binary solvent system of 60:40 water:acetonitrile (Spectrum Chemicals, New Brunswick, NJ, USA) with 10 mM ammonium acetate (Sigma-Aldrich, St. Louis, MO, USA) (A) and 90:10 isopropanol:acetonitrile (Spectrum Chemicals, New Brunswick, NJ, USA) with 10 mM ammonium acetate (B) at a flow rate of 0.4 mL/min using the following gradient conditions: linear gradient from 40–100% B (0–10 min), hold at 100% B (10–12 min), and then return to initial conditions and re-equilibrate (40% B; 13–15 min). An 8 μL sample was injected onto an Acquity HSS T3 column (2.1 × 100 mm, 1.8 μm; Waters Corp., Milford, MA, USA) maintained at 55 °C.

The column was eluted into the mass spectrometer fitted with an electrospray ionization (ESI) probe and analysis was performed in high resolution, MSE, positive, and negative modes with a scan time of 0.20 s and a mass scan range of 250–1800 *m*/*z*. Source conditions were: capillary voltage 2.4 (negative mode) and 3.2 (positive mode), temperature 120 °C, cone voltage 30, source offset 80, desolvation temperature 350 °C, cone gas 50 L/h, desolvation gas 500 L/h, and nebulizer gas 6.0 bar. For accurate mass, leucine–enkephalin (Waters Corp., Milford, MA, USA) at a concentration of 200 ng/mL was continually infused through the reference sprayer at five μL/min.

The lipidomic data files were processed with MarkerLynx XS application manager software (version 4.1; Waters Corp., Milford, MA, USA) for unsupervised feature discovery, principal component analysis (PCA), and orthogonal partial least-squares discriminant analysis (OPLS-DA), with the following parameters: mass tolerance of 0.02 Da, Xic window of 0.02 Da, mass window of 0.02 Da, retention time window 0.2 min, marker intensity threshold of 4000 intensity units, and data deisotoped. MarkerLynx XS features were normalized to dry mass of shoot tissue. A principle component analysis (PCA) and partial least-squares discriminant analysis (PLS-DA; Appendix A) were performed on the data using the statistical analysis module of the web program Metaboanalyst [59]. Feature peak area values were generated from the average of 3 technical replicates for each of the 6 biological replicates (Appendix A). From the 182 features detected in negative mode, 68 unique lipid species were identified. Positive mode was used to identify unique TAG species, with 8 TAG species identified from 166 detected features. Fragmentation analysis by MS/MS was performed to assign identities to the features using the same chromatography and source conditions described above. Lipid species were identified by hand using fragmentation patterns of head groups and acyl chains, using LipidMaps and previously published lipid datasets as references [58,60,61,62,63].

### 4.11. Lipidomics Statistical Analysis

For lipidomic analysis, tissue was grown according to randomized fully orthogonal blocked design, with six biological replicates of each genotype and Pi treatment. Each biological replicate was analyzed in technical triplicate by ESI-MS/MS. Peak area data were cube root-transformed to fit a Gaussian distribution. JMP Pro v16 (SAS Institute Inc., Cary, NC, USA) was used to perform linear regression modeling and backward model selection for all main effects and their interactions. A post hoc Tukey HSD test was performed to determine significance (α = 0.05) between genotypes and Pi treatments.

## 5. Conclusions

Together, the data presented here highlight the complexity of Pi homeostasis regulation in plants and suggest that InsPs as well as multiple PP-InsPs could contribute to a redundant system of control over PSR. Our results illustrate that perturbations to PP-InsPs influence a diverse set of processes including root morphology, gene expression, metabolite levels, and lipid remodeling events, in response to changing and/or limiting Pi conditions. Given that agricultural sources of Pi are predicted to be limited in the future, understanding the impact of plant cellular PP-InsPs on the biology and biochemistry of Pi-sensing is a critical endeavor.

## Figures and Tables

**Figure 1 metabolites-11-00601-f001:**
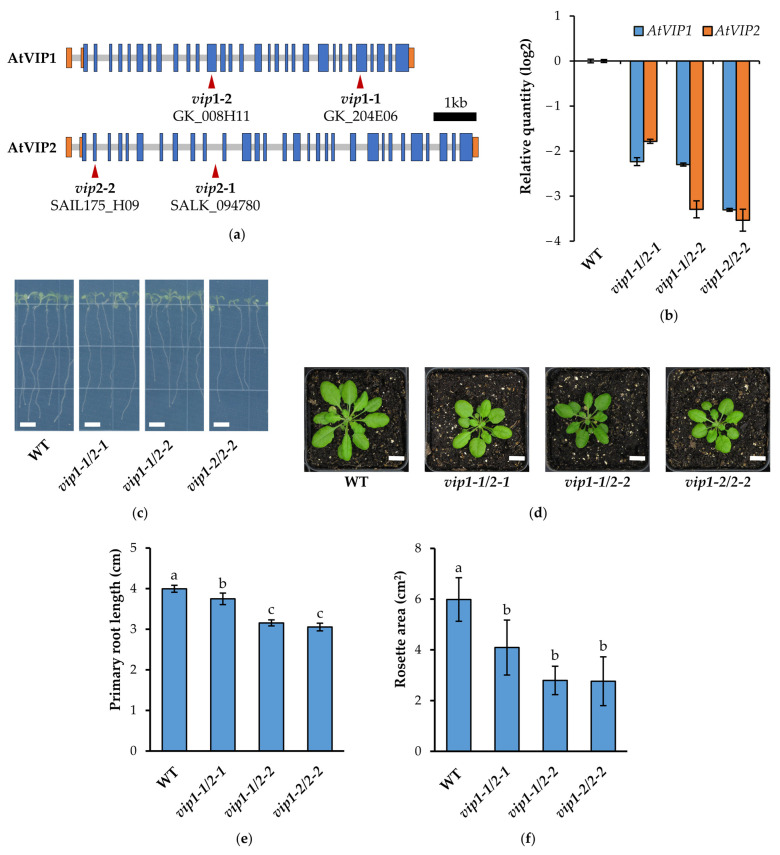
Double *vip* mutant generation and morphometric analysis. (**a**) Gene structure of *Arabidopsis AtVIP1* (AT3G01310) and *AtVIP2* (AT5G15070). Exon coding regions are indicated in blue, untranslated regions are indicated in orange, and introns are indicated in gray. The position of the T-DNA insertions are indicated by the red arrows. Three independent homozygous double mutants (*vip1-1*/*vip2-1*, *vip1-1*/*vip2-2*, *vip1-2*/*vip2-2*) were generated by crossing. (**b**) *AtVIP1* and *AtVIP2* transcriptional abundances in 14-day-old seedlings relative to WT. Samples were assayed in triplicate by qRT-PCR using gene-specific primers and *PP2A* as an endogenous control. Error bars denote SD. (**c**) WT and *vip* double mutant seedlings were grown on 0.5× MS media for 10 days (scale bars indicate 0.5 cm). (**d**) Young seedlings were transplanted to soil and grown under short day conditions (8 h light) and rosette area was measured after 28 days (scale bars indicate 1 cm). (**e**) Primary root lengths. Data presented are the average ± SD of two independent experiments (*n* = 50/experiment). Different letters indicate statistically significant means (Tukey HSD, α = 0.05). (**f**) Rosette areas. Data presented are the average ± SD of a representative experiment (*n* = 12). Different letters indicate statistically significant means (Tukey HSD, α = 0.05).

**Figure 2 metabolites-11-00601-f002:**
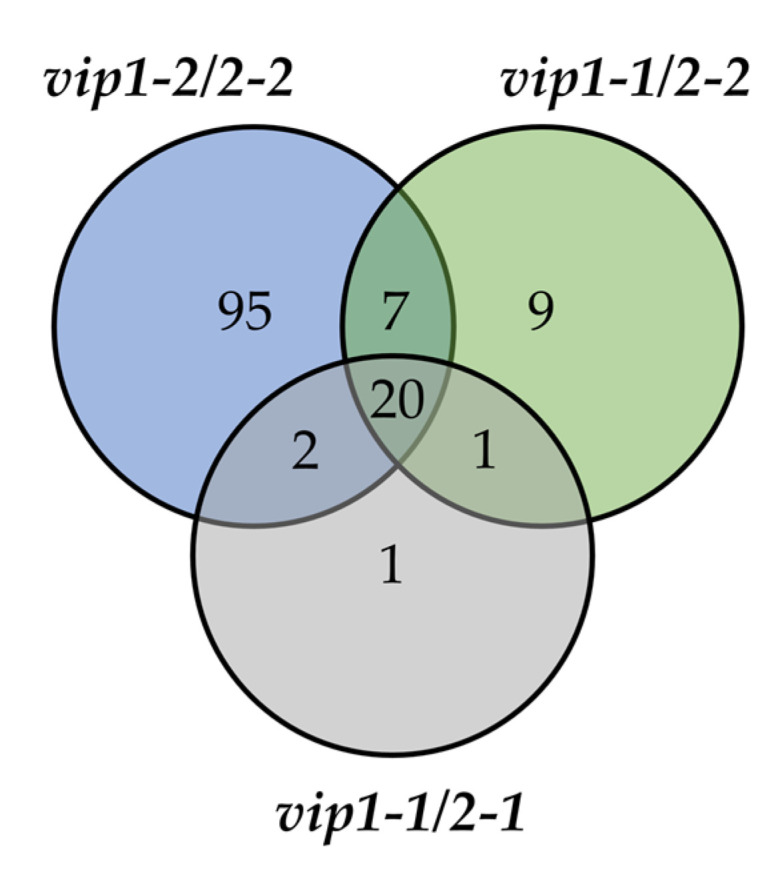
Genes downregulated in shoot tissue of *vip* double mutants. Venn diagram shows overlap between genes downregulated in all three *vip* mutants compared to WT. A conserved subset of 20 genes were identified as being downregulated in all the *vip* double mutants.

**Figure 3 metabolites-11-00601-f003:**
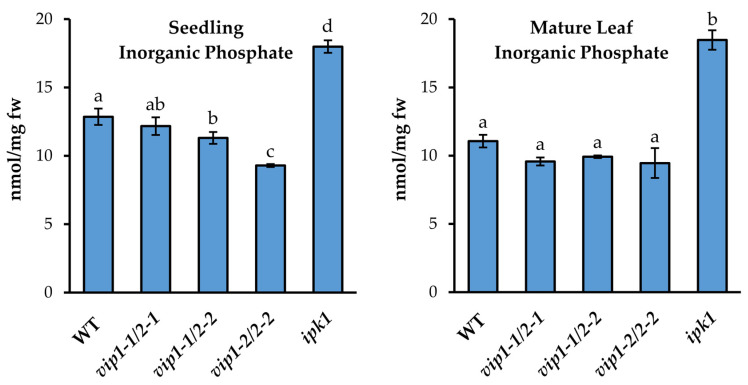
Inorganic phosphate content was quantified in 18-day-old seedlings and in 49-day-old mature leaves of WT, *vip* double mutants, and *ipk1*. Data presented are the average ± SD of three independent bioreplicate experiments. Different letters indicate statistically significant means (Tukey HSD, α = 0.05).

**Figure 4 metabolites-11-00601-f004:**
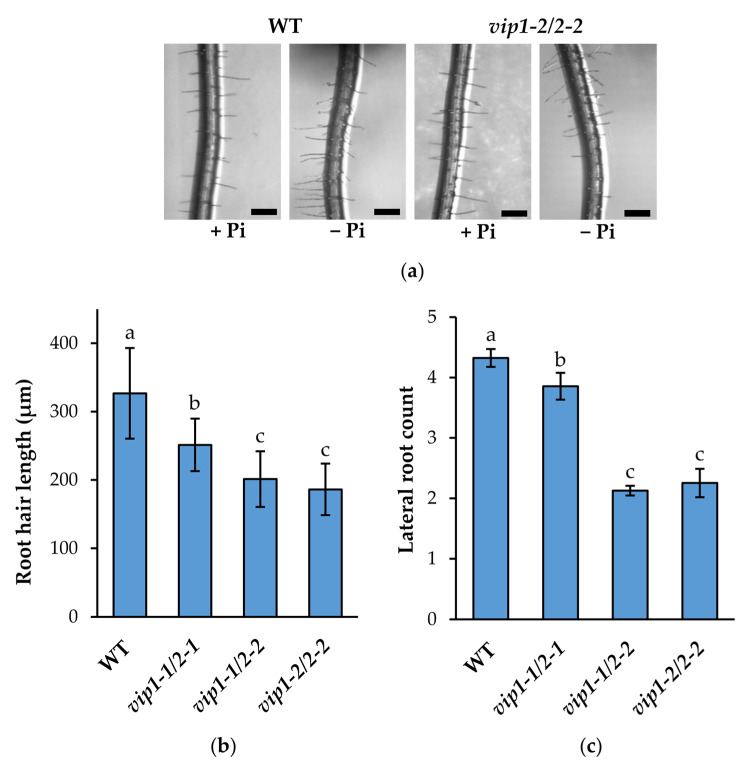
Root system architecture (RSA) responses in *vip* mutants. (**a**) Close-up images of 14-day-old WT and *vip1-2*/*vip2-2* primary roots from seedlings grown on phosphate-replete (+Pi) and phosphate-limiting (−Pi) MS media (scale bars indicate 250 µm). (**b**) Root hair length of seedlings grown on phosphate-limiting media. Data presented are the average ± SD of two replicate experiments, (*n* = 400–500 observations). (**c**) Lateral root count of seedlings grown on phosphate-limiting media. Data presented are the average ± SD of three bioreplicate experiments (*n* = 30 observations/bioreplicate). Different letters indicate statistically significant means (Tukey HSD, α = 0.05).

**Figure 5 metabolites-11-00601-f005:**
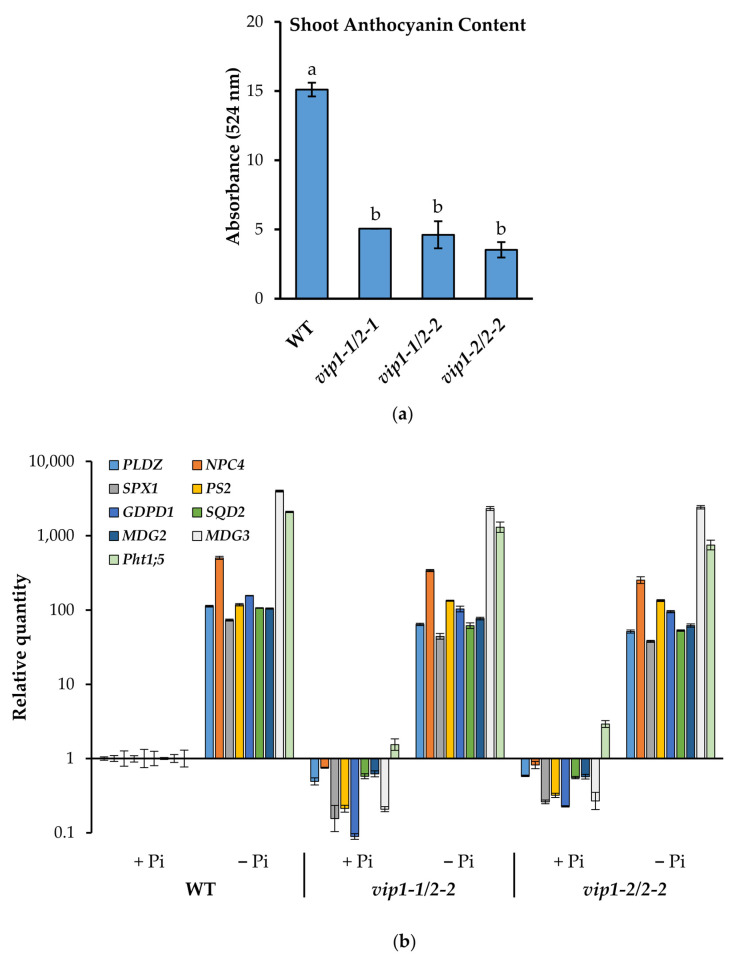
Shoot transcriptional responses to limiting Pi conditions. (**a**) Anthocyanin accumulation in 18-day-old WT and *vip* double mutant seedlings grown on phosphate-limiting media. Data presented are the average ± SD of two bioreplicate experiments assayed in technical duplicate. Different letters indicate statistically significant means (Tukey HSD, α = 0.05). (**b**) The transcriptional abundance of select PSR genes was queried relative to WT by qRT-PCR using gene-specific primers and *PEX4* (which is not transcriptionally responsive to low Pi) as an endogenous control. Data shown are from a representative bioreplicate. Error bars indicate standard deviation.

**Figure 6 metabolites-11-00601-f006:**
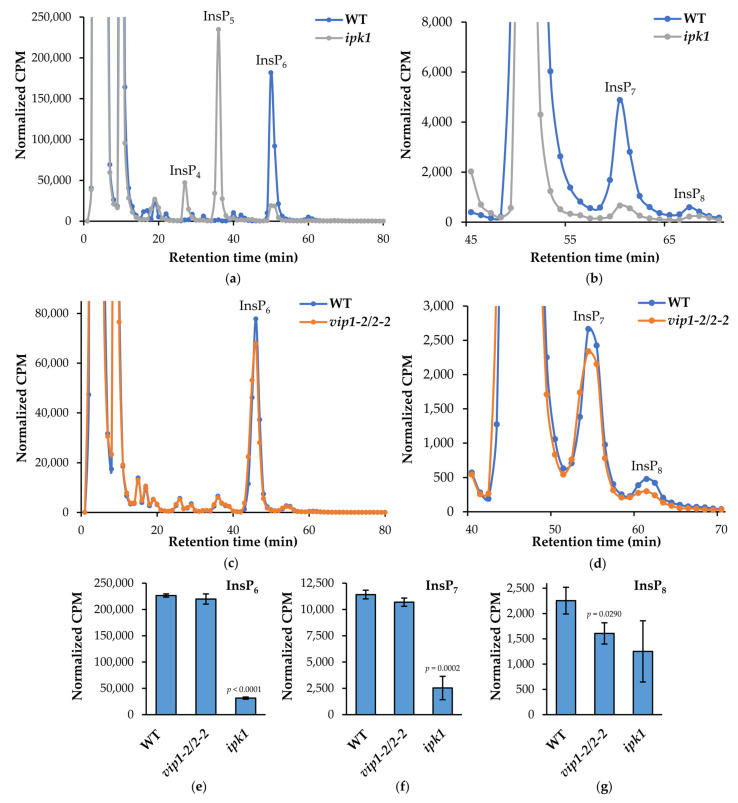
Inositol phosphate profiles. Fourteen-day-old WT, *vip1-2*/*vip2-2* and *ipk1* seedlings were radiolabeled with ^3^H-*myo-*inositol for 5 days and InsP species were separated by HPLC and quantified by liquid scintillation counting. (**a**–**d**) Normalized CPM elution profiles comparing WT to *ipk1* (**a**,**b**) and *vip1-2*/*vip2-2* (**c**,**d**). (**e**–**g**) Normalized CPM from fractions comprising InsP_6_ (**e**), InsP_7_ (**f**), and InsP_8_ (**g**). Mean values of three replicate experiments are shown (*n* = 3) and error bars represent standard deviation. Significance of differences relative to WT were calculated by two tailed Student’s *t*-test.

**Figure 7 metabolites-11-00601-f007:**
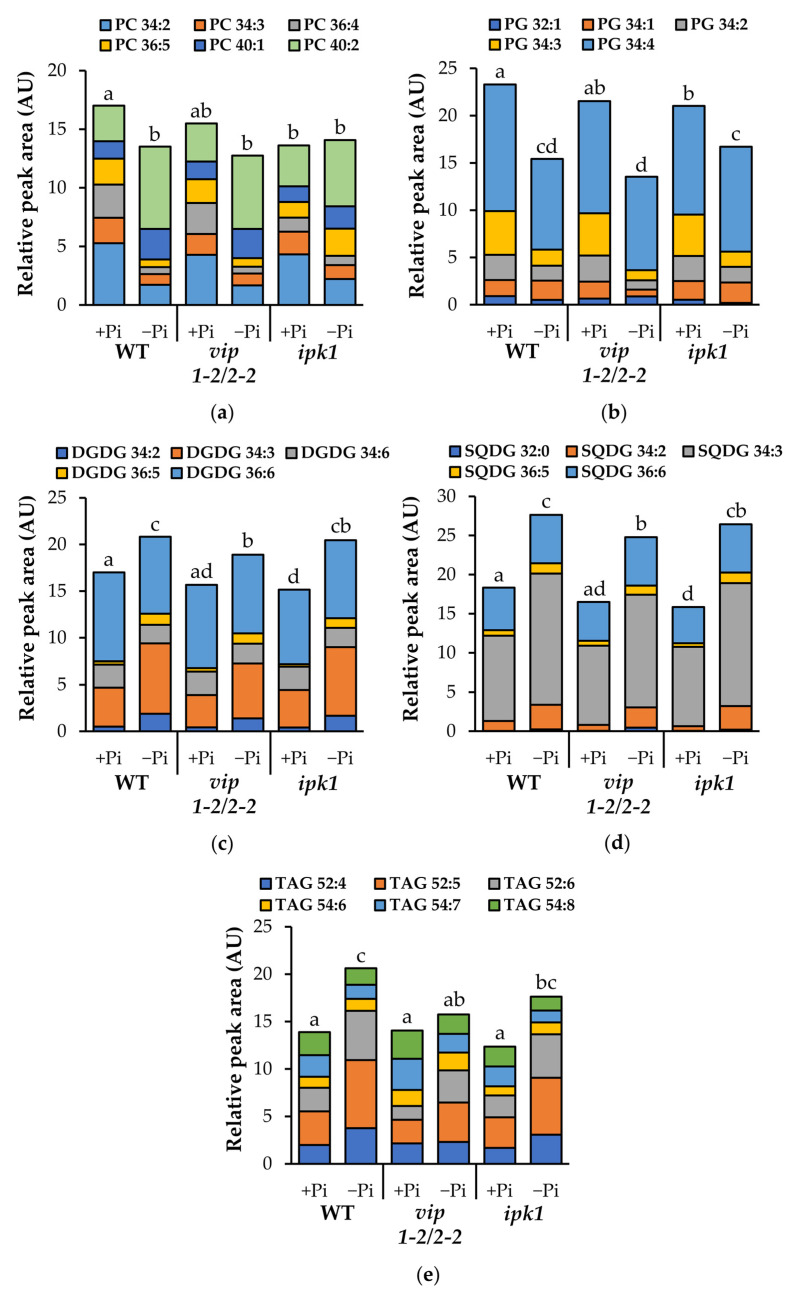
Composition of major phospholipid and galactolipids under Pi-replete and limiting conditions. Lipids were extracted and analyzed from wild type (WT), *vip1-2*/*2-2*, and *ipk1* in 18-day-old shoot tissue grown on 1 mM Pi (+Pi) and 10 µM Pi (−Pi) agar media. Individual lipid species are indicated by stacked bars. (**a**) Relative peak area of phosphatidylcholine (PC). (**b**) Relative peak area of phosphatidylglycerol (PG). (**c**) Relative peak area of digalactosyldiacylglycerol (DGDG). (**d**) Relative peak area of sulfoquinovosyldiacylglycerol (SQDG). (**e**) Relative peak area of triacylglycerol (TAG). Values are cube root-transformed relative peak area units and means of six biological replicate experiments assayed in technical triplicate. Letters (**a**–**e**) indicate statistical groups according to post hoc Tukey HSD test (α = 0.05).

**Table 1 metabolites-11-00601-t001:** Description of conserved downregulated genes in *vip* double mutants. Many of these genes were found to be associated with phosphate starvation responses. Promoter motif analysis found one or more PHR1-binding sequences (P1BS, GnATATnC) to be present in the 1kB putative promoter region upstream of 14 of these genes.

Locus ID	Gene Description (Symbol)	−1 kb P1BS
AT1G17710	Phosphoethanolamine/Phosphocholine Phosphatase 1 (*PEPC1*)	2
AT1G23110	Fold Protein	1
AT1G58280	Phosphoglycerate Mutase Family Protein	1
AT1G73010	Phosphate Starvation-induced Gene 2 (*PS2*)	2
AT2G11810	Monogalactosyldiacylglycerol Synthase Type C (*MGDC*)	0
AT2G46880	Purple Acid Phosphatase 14 (*PAP14*)	2
AT3G01345	Expressed Protein	0
AT3G02040	Glycerophosphodiester Phosphodiesterase 1 (*GDPD1*)	4
AT3G05630	Phospholipase D Zeta 2 (*PLDZ2*)	4
AT3G09922	Induced By Phosphate Starvation1 (*IPS1*)	2
AT3G17790	Purple Acid Phosphatase 17 (*PAP17*)	0
AT3G30122	Pseudogene	1
AT3G43110	Transmembrane Protein	1
AT4G33030	Sulfoquinovosyldiacylglycerol 1 (*SQD1*)	0
AT5G03545	Induced By Pi Starvation 2 (*ATIPS2*, *AT4*)	2
AT5G15070	*Arabidopsis* Homolog Of Yeast Vip1 2 (*ATVIP2*)	0
AT5G20150	Spx Domain Gene 1 (*SPX1*)	1
AT5G20410	Monogalactosyldiacylglycerol Synthase 2 (*MGD2*)	2
AT5G20790	Transmembrane Protein	0
AT5G53048	Natural Antisense Transcripts	1

## Data Availability

Next-generation sequencing data reported in this study will be deposited to the National Center for Biotechnology Information Sequence Read Archive (SRA) database available online at: https://www.ncbi.nlm.nih.gov/sra/ (accessed on 25 March 2021). Data were compared to a published dataset (Dong et al., 2019) available online at: https://www.cell.com/cms/10.1016/j.molp.2019.08.002/attachment/02fad813-965e-4a57-aa06-1150424ecf0b/mmc2.xlsx (accessed on 25 March 2021).

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
