# Peer review of "A Role for Inositol Pyrophosphates in the Metabolic Adaptations to Low Phosphate in Arabidopsis"

_metabolites, 2021, doi:10.3390/metabo11090601_

Round 1

Reviewer 1 Report

This manuscript used double mutant of two VIP genes in InsP8 synthesis to show the effect of P singling on metabolism adaptation of Arabidopsis to P limitation. It provides an interesting regulative effect with solid evidence at physiological, morphological, biological and transcriptional levels. However, its writing is poor and should be fully re-written before further consideration. Major concerns. 1. There are a lot redundant description. For example, anything clearly mentioned in Introduction should not repeat in Results, same conditions should not repeatedly descript again and again in M&M… 2. The description for lipids (Result 2.8) is disorder and unconcise. I would suggest making points one by one, addressing the patterns under normal growth condition first and then the pattern changes under low Pi, to compare WT with VIP mutants first and VIP with ipk mutants second. 3. Discussion should be deep-thought and be in the way to form the major conclusion expressed by title. A working model will be helpful for readers to understand the whole picture. Minor concerns. 1. Figure 4 b and c. Were they under P limitation? 2. Word “Arabidopsis” should be in italic throughout all text. 3. The style of all results should be same. 4. What is the difference between “leaf” and “shoot” during harvesting, considering Arabidopsis is rosette plants.

Reviewer 2 Report

The manuscript is well written and therefore it can be accepted for publication just after a minor spell check I congratulate the authors for their nice work.

Author Response

Thank you for the kind comments.

Reviewer 3 Report

Comments and Suggestions for Authors

The manuscript titled “Metabolic Adaptations to Low Phosphate in Plants; A Role for Inositol Pyrophosphates” Manuscript ID: metabolites-1312943, by Land et al. provides insight on the complex regulatory mechanisms underpinning phosphate homeostasis in plants in general and in the context of phosphate starvation particularly. The presented study is a very interesting piece of research that proofs the response to PP-InsPs influence a diverse set of processes including gene expression, metabolite levels, and lipid remodeling events, in response to changing and/or limiting Pi conditions. In the current study, the authors used comprehensive T-DNA insertion collection and a generated vip1/vip2 Double mutants to characterize the response of the model plant, Arabidopsis thaliana, to low phosphate stress. Moreover, they used multiple techniques included but were not limited to, NextGen RNA sequencing, plant inositol polyphosphate analysis, lipidomics, …etc. to prove that evidence that the vip1/vip2 mutants are less responsive to low phosphate compared to wild type plants. It is worth mentioning that idea is interesting, the study is well-planned, and logical linking between different sections is obvious. However, I have some comments and suggestions for authors just to improve the final version of the manuscript.

Major concerns

Firstly, my main concern is about statistical analysis and data curation.

  1. The statistical analysis is poorly described throughout the manuscript. As I understand from the current study, the authors testing different treatments in each experiment, without any explanation of the used experimental design or clarification for how they conducted this statistical analysis. The experimental design showed be described and clarified for each individual experiment. It is highly recommended to the authors that they should add a new section to the “materials and methods part” under the sub-title of “statistical analysis” to (1) describe the used experimental design in detail (Completely randomized design [CRD], randomized complete block [RCB], split-plots, or something else); (2) number of biological and technical replicates per treatment for each experiment; (3) describe statistical analysis they carried out (ANOVA, t-test, or something else), (4) which Post-Hoc test they used (Bonferroni Procedure, Duncan’s MRT, Dunn’s MCT, Tukey’s HSD test, or something else). This issue must be addressed by the authors in the next revised manuscript. Moreover, I could recommend redoing the statistical analysis and you should present the statistical differences using significance letters, instead of asterisks (** or ***) and include the calculated p-value within each figure. I understand that some of the statistical analyses were briefly described within the results section or figures caption, however, I believe that this is not the appropriate place for that. This issue is critical and should be fixed carefully and clearly in the revised version of this manuscript.
  2. On page 11 (lines 313-314): within figure 7 caption, the authors mentioned that “Values are relative peak area units and means of two biological replicate experiments assayed in technical triplicate. Letters (a-e) indicate statistical groups according to pairwise t-tests”. And on page 18 (lines 667-668): the authors mentioned that “Averaged feature peak area values were generated from the average of 3 technical replicates of 2 biological replicates”. As I understand here, the authors add together technical replicates (3 technical replicates) and biological replicates (2 biological replicates) which boosts their n value from 2 to 6. I find fault with this method of calculating “n” and I would not recommend this paper for publication until this method has been either (1) better described/supported or (2) fixed/re-calculated. If the analyses were conducted soundly, but the description is wrong, this is an easy fix. However, a reduction in n-value may reduce the significance of certain statistical results since using only TWO biological replicates is not valid for statistical analysis. Therefore, so this matter is extremely important. Technical replicates should not count towards the n-value. Technical replicates are solely to verify the reproducibility and variability of their analytical method, and they are not to be used for the higher level, statistical analyses to draw biological conclusions. Only biological replicates are to be used for this purpose. Using technical replicates as pseudo-biological replicates (pseudoreplication) is falsely boosting the statistical power of any results obtained from these samples. If more statistical power is needed, then more biological replicates are collected. The way the authors described their statistical analysis looks like they are grasping for more statistical significance than they actually have. I am truly curious to learn more about this method and what the authors actually did. Take, for example, a study that seeks to measure the average plant height in your experiment. If you measure 3 plants (3 biological replicates) ten times (technical replicates) and compute their average, this will give a very different result than measuring thirty different plants (30 biological replicates), especially if those three initial plants are either all tall or all short. This issue must be addressed by the authors before the paper can be considered for publication because it has impacts on their results and interpretation.
  3. On page 11 (line 314): Again, within figure 7 caption, the authors mentioned that “Letters (a-e) indicate statistical groups according to pairwise t-tests”. Although the authors did not mention the used experimental design (they should do in the next revised manuscript), I can tell from the figure construction that they used a split-plot design with the Arabidopsis mutants as the main factor (WT, vip1-2/2-2, and ipk1) and the Pi levels as a sub-factor (1 mM [+Pi] vs. 10 µM Pi [-Pi]). However, the presented statistical analysis (using t.test as they mentioned) does not reflect that. I could recommend redoing the statistical analysis according to the split-plot design and presenting the p-values within the figure. Briefly, according to the statistical analysis of split-plot design, you will get three p-values; p mutants to compare between Arabidopsis mutants regardless of the Pi level, p Pi levels to compare between both Pi treatments regardless of the Arabidopsis mutants, and p mutants×Pi levels to compare the interaction and you should present the statistical letters based on the latest p-value (p mutants×Pi levels).
  4. Finally, Although the language used in the manuscript is easy to follow and understand, however, the manuscript should be carefully and deeply revised for grammar and English use, since some other mistakes were found throughout the whole paper.

Other comments and suggestions

  • Title
  • According to the “Instructions for Authors” of Metabolites, the title of your manuscript should be concise, specific, and relevant. It should identify if the study reports (human or animal) trial data, or is a systematic review, meta-analysis, or replication study. When gene or protein names are included, the abbreviated name rather than full name should be used. Therefore, I suggest paraphrasing the title and make it straightforward. Suggested title “Role of Inositol Pyrophosphates in Arabidopsis Adaptations to Low Phosphate conditions”. For more details about this issue, please see the “Instructions for Authors” (https://www.mdpi.com/journal/metabolites/instructions).
  • Abstract
  • Although the abstract is easy to read, the abstract is very descriptive. About two-thirds of the abstract is an introductory statement (lines 9-17) and does not present the main findings of the work. Again, according to the “Instructions for Authors” of Metabolites, the abstract should be a total of about 200 words maximum. The abstract should be a single paragraph and should follow the style of structured abstracts but without headings: 1) Background: Place the question addressed in a broad context and highlight the purpose of the study; 2) Methods: Describe briefly the main methods or treatments applied. Include any relevant preregistration numbers, and species and strains of any animals used. 3) Results: Summarize the article's main findings, and 4) Conclusion: Indicate the main conclusions or interpretations. The abstract should be an objective representation of the article: it must not contain results that are not presented and substantiated in the main text and should not exaggerate the main conclusions. Therefore, I would recommend rewriting the abstract to articulate more clearly the main findings of the work. It needs to be more informative rather than descriptive. Also, remove the long introductory statement from the beginning of the abstract and rewrite it straightforward to ensure clarity. For more details about this issue, please see (https://www.mdpi.com/journal/metabolites/instructions).  
  • All abbreviations should be defined in parentheses the first time they appear in the abstract. Please spell out all abbreviations (such as PHR1, VIP, IPK1, and more.) at their first appearance in the abstract.
  • Introduction
  • All abbreviations should be defined in parentheses the first time they appear in the main text, and figure or table captions. For instance, MYB (Line 64) SYG1/PHO81/SPR1 (Line 68), VIP or VIH (Line 82), and so many more, should be spelled out at their first appearance. This issue should be carefully revised throughout the manuscript. For example:
  • Results
  • Line 127-129: the authors mentioned that “Furthermore, RT-PCR amplification of cDNA derived from these double mutants did not detect full-length VIP1 or VIP2 transcripts indicating an absence of functional full-length VIP1 and VIP2 proteins (data not shown)”. Why are these data not shown? Please include these data within the main text or with the supplementary materials.
  • The figure presentation needs to be enhanced before publishing. For example:-
  • For multi-panel figures, please use bold, 14 pt, upper case letters (A, B, C, …etc) to identify each panel. Panel label identifications should appear in the upper left corner of each panel, not at the bottom as presented within your figures.
  • Capitalize only the first word and proper nouns in each label on the x and y-axis.
  • Scale bars within figure 1C, figure 1D and Figure 4A should be inserted to indicate magnification on the figure itself not only within the caption.
  • Throughout the manuscript, the presented data is NOT adequately analyzed in terms of statistical analysis. The asterisks (** and ***) should be replaced by lower-case letters (a, b, c, …etc) for significant purposes in all figures. Appropriate statistical analysis and significant letters should be presented within all graphs, along with p-values.

Round 2

Reviewer 1 Report

Comments and Suggestions for Authors

I see significant improvements in the revised manuscript. However, the writing is not satisfied for publish yet. I also focus on scientific issues in current round review and list them below.

Major concerns.

  1. I would like to see specific and informative points, not general description in Abstract. By the way, “this study …” and “In this study” are closely together.
  2. I think the genes/mutants are key information in the paper, they should be included in keywords.
  3. line 260-262. “subset of downregulated genes in the vip double mutants are likely under the control of the PHR1 transcription factor”. Is this regulation for the genes previously reported or a novel finding?
  4. line 345-358. What is needed here is a conclusive sentence for the presented data, not a long discussion.
  5. line 250-251 and 364. Term “lipid remodeling genes”, do they specifically function in lipid remodeling, or function in lipid metabolism and consequently remodel lipids under certain conditions?
  6. line 422, I am confused by “Altered Lipid Remodeling Responses in Mutants Altered in PP-InsPs”.
  7. Fig. 7a top panel, PC levels between +P and -P in ipk1 are same. Why is that and does it support author’s conclusions in line 467 and 470-472?
  8. Why Conclusion does not follow the Discussion? The two sections are separated by Materials and Methods, is this good for reading or required by the journal?
  9. Fig. 7. Author arranged Peak Area at top and Relative Peak Area at bottom for same lipid class in a, c, d, and e. Why not in b?  

Minor concerns.

  1. line 48, what does the “changes” refer to?
  2. linewidth should be uniform in whole text. Otherwise reading will be uncomfortable.
  3. Fig. 1b, half labels?
  4. Fig. 5a, “Absorbance (524 nm)” will be better.
  5. Fig. 6. Can the statistical labels be uniform in the figures with letters?
